# A Novel High-Fidelity Simulation for Finishing Operations: Hybrid Image Mosaic and Wavelet Decomposition

**DOI:** 10.3390/mi15070834

**Published:** 2024-06-27

**Authors:** Yupeng Xin, Wenhui Li, Xun Xu, David Culler

**Affiliations:** 1College of Mechanical and Vehicle Engineering, Taiyuan University of Technology, Yingze Street, Taiyuan 030024, China; 2College of Aeronautics and Astronautics, Taiyuan University of Technology, Daxue Street, Taiyuan 030600, China; liwenhui@tyut.edu.cn; 3Department of Mechanical and Mechatronics Engineering, The University of Auckland, Grafton Road, Auckland City 1010, New Zealand; x.xu@auckland.ac.nz; 4Manufacturing and Mechanical Engineering and Technology, Oregon Institute of Technology, Klamath Falls, 3201 Campus Drive, Klamath, OR 97601, USA; david_culler@yahoo.com

**Keywords:** discrete simulation, micro-surface models, finishing, image processing

## Abstract

In finishing simulations, achieving accurate results can be challenging due to the minimal amount of material removal and the limited measurement range of surface micro-topography instruments. To overcome these limitations, a novel high-fidelity modeling method combining image mosaic and wavelet decomposition technologies is proposed in this paper. We achieve the stitching of narrow field and high pixel micro morphology images through four steps: image feature extraction, overlapped feature matching, feature fusion, and stitching effect evaluation. On this basis, the wavelet decomposition method is employed to separate detection signals based on their respective frequencies, allowing the establishment of a datum plane and a roughness surface. The point cloud model undergoes a transformation into a continuous geometric model via the Poisson reconstruction algorithm. In the case study, four sample images of an aluminum alloy sheet after barrel finishing were collected using the ZeGage Plus optical profiler. Each image has an actual size of 834.37 μm × 834.37 μm. Subsequently, a comparison was carried out between the physical and simulation experiments. The results clearly indicate that the proposed method has the potential to enhance the accuracy of the finishing simulation by over 30%. The error between the resulting model and the actual surface of the part can be controlled within 1 μm.

## 1. Introduction

There is a growing trend toward digitalization in manufacturing that allows people to replace machining trials with computer simulation technology. The potential to reduce production costs, eliminate waste, and improve efficiency has attracted the attention of researchers and industry [1]. While early applications of machining simulations focused on optimizing tool paths and predicting tool/holder collisions, the technology has expanded into other areas. With advances in Finite Element Modelling (FEM), there are opportunities for new and innovative metal cutting simulations to be developed [2]. The deformation process of plastic metal chips and the change in cutting surface temperature and pressure can be simulated using FEM. Applying this method, the cutting action is modeled as a Boolean operation, providing moderate simulation times and both relevant and fairly accurate surface topographies [3].

However, the simple Boolean intersection assumption is not directly applicable for the simulation of finishing operations because of the tiny amount of material removed and the lack of a micro-surface topography. In this case, a significant number of examples from the literature represent the finishing process as the cinematic mapping of the tools or particles on the smooth surface [3,4,5,6]. By doing that, the surface shape of the part on the micro scale is ignored. The simulation results based on such an ideal model often differ greatly from reality [7]. Therefore, constructing a high-fidelity surface part model that can be directly applied in the finishing simulation is critical to improve the accuracy and benefits of simulation [6].

The ISO standard 25178 and ASME B46.1 defines a uniform description for surface models using surface texture (surface roughness, waviness, and lay) [8]. An ideal surface model, also known as a Nominal Model, is a continuous surface composed of an infinite number of points on the geometric shape of parts. Subtractive machining processes like milling, turning, and grinding are imperfect by nature, and therefore produce imperfections and deviations in part surfaces such as pits and bumps. A high-fidelity surface model should map these machining defects into the 3D model. There are two distinct strategies for simulating machining deviations on part surfaces.

The CAE-based machining simulation strategy, which simulates the detailed machining process and generates high-fidelity models (known as Schleich) divides geometric deviations into system deviation and random deviation according to GPS standards. Then, it utilizes geometric modeling methods, such as Bezier curves [9,10], Splines [11,12,13], and NURBS [14,15], to establish the system deviation model. The Gaussian process method is used to establish the random deviation model, thereby ensuring that a high-fidelity part surface model is transferred to the CAE system [16,17,18]. The simulation results obtained by this strategy are based on the premise that the ideal processing state remains unchanged and does not include the actual physical state.

To address this deficiency, some researchers study the use of the machine learning method to train high-fidelity models from historical machining data. Jose proposed a reconstruction algorithm that takes as input an incomplete measurement, identifies the statistical shape parameters, and outputs a full scan reconstruction [19]. Nicholas utilized machine learning in the ultra-precision diamond machining of single-crystal germanium to create a model, which demonstrated that by using surface metrology parameters in the model, improvements in fracture prediction can be achieved [20]. Gaikwad used machine learning techniques to train high-fidelity models from historical monitoring data obtained through high-speed imaging, for predicting the size, velocity, and shape characteristics of droplets in the liquid metal jetting additive manufacturing process [21]. The trained high-fidelity models can effectively improve prediction results when processing conditions remain constant.

The second strategy is based on real-time physical detection data. The detected physical signals are converted into numerical values using mathematical statistical methods, and then surface and solid models are generated using geometric modeling techniques. Min uses one-dimensional and three-dimensional random noise to generate the part surface model with shape deviation [22], but because the vertex coordinates of the surface model simulated by random deviation are generated independently, the distance between the two connected vertices may be too large, resulting in a surface that is not smooth. Rui Xiong used a full-dimensional measurement method based on polarized coded aperture correlation holography, which greatly improves the reliability and practicability with respect to conventional measurement methods [23]. Fireman and Xun developed a surface roughness predictive model by obtaining machining parameters, e.g., feed rate, depth of cut, and spindle speed [24].

In recent years, digital twin technology has gained significant traction, promoting an increasing number of scholars to develop digital twin systems that enable the creation of high-fidelity models. Bao proposed a biomimicry-based digital twin modeling method to construct high-fidelity models by fusing machining process data and optimizing the machining process in real-time [25]. Xin proposed a refined simulation method based on digital twin technology, which integrates real-time machining data with a process design system through the integration of CAPP and MES systems. The high-fidelity process model is iteratively generated using real-time inspection data to improve the fidelity of process simulation [26]. Similar methods have also been applied to cutting tools [27,28], machine tools [27,29,30], and process models [31,32,33]. Admittedly, the modeling methods based on the actual detection data are more conducive to improving the overall accuracy of the model.

In practical applications, the accuracy of the aforementioned high-fidelity geometric models is often influenced by various factors such as model complexity, computer performance, and data transmission. In addition, data collected through optical measurement is often affected by environmental interference, equipment errors, and other factors, requiring the filtering and processing of the data [34]. Mean and median filtering are two commonly used basic filtering methods that reduce the influence of outliers by calculating the average and median values of the data, respectively. Liu used median filtering to denoise the vibration signals during the milling of thin-walled workpieces [35]. Si implemented Kalman filtering to estimate the state of the engine support system by establishing a system state model and an observation model [36]. When the signal is extremely unstable, it is usually necessary to adjust the filtering parameters in real-time based on the change or noise characteristics. This type of filtering method is called adaptive filtering. He proposed an adaptive filtering method based on singular values and a mixture of Gaussian models, which accurately identified the tool wear state under different machining parameters [37]. The wavelet transform method is a time–frequency analysis method that decomposes a signal into multiple sub-signals of different frequencies and filters each sub-signal. Its advantage is that it can simultaneously process high-frequency and low-frequency signals. Ahmed used the wavelet transform method to monitor the changes in cutting force signals in stainless steel machining, achieving effective control of the built-up edge [38].

Based on the above research, it is clear that the geometric accuracy of high-fidelity models generated by different filtering methods may vary, making it challenging to ensure consistency. Furthermore, detection signals reflect various geometric morphologies at different scales. Surface roughness and waviness are associated with distinct frequencies and wavelengths, as illustrated in Figure 1. The v/div and s/div refer to the time base unit of the oscilloscope, representing the volt value per grid and seconds per grid, respectively. Therefore, separating, extracting, and processing these signals will significantly affect the accuracy of the surface model.

To address the shortcomings described above, this paper proposes a high-fidelity surface modeling method that combines an image mosaic technology with wavelet decomposition. A flowchart to describe this process is shown in Figure 2.

Following this logic, the paper is structured as follows: Section 2 describes the concept of the image mosaic, Section 3 compares the three different wavelet decomposition methods and describes how the Symlet wavelet was selected to reconstruct the high-fidelity surface model, and Section 4 compares and analyzes the results of the simulation developed and the finishing experiment employed. Finally, conclusions and some areas identified for future research are described.

## 2. Image Mosaic

The CCD camera imaging system is typically used to image the micro-shape of a surface measured by the 3D topography instrument [39,40]. But, due to the restriction of the lens, the transverse dimension of the measured surface is limited to approximately a 1000 μm × 1000 μm viewing area [32]. When measuring a large area with a high-power lens, it is necessary to expand the field of view of the measured surface. The work presented in this paper employs a method for dividing the surface of the test part into several overlapping sub-areas for the purpose of measuring. The algorithm is designed to cut and splice the overlapping areas and create a larger range of 3D surface topography data.

Using a 2 × 2 image to demonstrate the stitching process, the data acquisition and processing diagram is shown in Figure 3. The splicing steps are as follows: (a) Move the lens to the first detection position to obtain the image *S*_1_. (b) Now, move the lens horizontally in the +x direction to the next detection position *S*_2_. Here, it is important that the horizontal distance *d*_1_ is less than the length of *S*_1_ so that images *S*_1_ and *S*_2_ are overlapped. (c) Next, move the lens back in the *−x* direction over *S*_12_ to obtain the area of calculating coincidence, being sure to completely cover the *S*_12_ overlapping area. (d) Repeat the same principle explained above to obtain the overlapping area of *S*_3_ and *S*_4_. (e) Lastly, complete the splicing of *S*_12_ and *S*_34_.

To demonstrate the methods proposed in this paper, a 20 mm × 20 mm × 3 mm aluminum alloy sheet is used. The part’s surface is measured with the ZeGage Plus optical profiler, which has sub-nanometer precision and an optical resolution of 0.52 μm. The sampling area selected for use is in the geometric center of the part. By using three equally spaced distances for moving the lens *d*_1_, a set of mosaic images with different coincidences are collected. The overlapping areas are about 10%, 20%, and 30% of the original image size. Figure 4 shows the image with a coincidence of 10%. The sampling interval is approximately 0.815 μm. In Figure 4, the image is enlarged by 700% and the result is 1200 × 1200 pixels. The original size of the composite image created using the stitching/splicing process described earlier is 834.37 μm × 834.37 μm.

### 2.1. Image Feature Extraction

The first step in the image mosaic process is to recognize the features within the overlapping regions. The most widely used feature description methods are the Harris corner algorithm [41] and the Scale-Invariant Feature Transform (SIFT) algorithm [42,43]. An analysis is performed to identify the optimal feature recognition by comparing the results from two algorithms.

(1)Harris corner feature extraction algorithm

A corner is defined as a stable sparse geometric feature in the point cloud data. The Harris algorithm finds corners by sliding the convolution window; when the sliding window moves at various angles, the corner points will be determined when the pixel gray value in the window changes [41]. The details are shown in Appendix A. Figure 5 shows the corner detection results of *S*_1_, *S*_2_, *S*_3_, and *S*_4_ in MATLAB 2020, which are 1561, 1551, 1578, and 1582.

(2)SIFT feature extraction algorithm

The SIFT algorithm includes four steps: scale space extreme value detection, key point location, direction assignment, and key point description [44]. The result of stitching an image using the SIFT algorithm in MATLAB is shown in Figure 6. The key point numbers of the four images (*S*_1_, *S*_2_, *S*_3_, and *S*_4_) are 1660, 1349, 1670, and 1548.

### 2.2. Image Feature Matching

After extracting the corresponding corner position of each image, the Normalized Cross Correlation (NCC) is used to match the feature points. Yoo and Han [45] demonstrate the robustness and accuracy of NCC, and its formula is as follows:(1)$$NCC=\sum_{\left(x,y\right)\in W}\left(I_{1}\left(x,y\right)-\bar{I_{1}\left(x,y\right)}\right)\cdot \left(I_{2}\left(x,y\right)-\bar{I_{2}\left(x,y\right)}\right)/\sqrt{\sum_{\left(x,y\right)\in W_{1}}{\left(I_{1}\left(x,y\right)-\bar{I_{1}\left(x,y\right)}\right)}^{2}}\cdot \sqrt{\sum_{\left(x,y\right)\in W_{2}}{\left(I_{2}\left(x,y\right)-\bar{I_{2}\left(x,y\right)}\right)}^{2}}$$
where *W* is the window of feature point. $I_{1}\left(x,y\right)$ and $I_{2}\left(x,y\right)$ are the pixel values in the original image. $\bar{I_{1}\left(x,y\right)}$ and $\bar{I_{2}\left(x,y\right)}$ are the average pixel values in the window.

The matching process consists of the follow steps: First, a point is selected in $I_{1}\left(x,y\right)$ and its corresponding feature point with the greatest correlation is searched for in $I_{2}\left(x,y\right)$. Then, for the selected point in $I_{2}\left(x,y\right)$, the feature point with the greatest correlation is searched for in $I_{1}\left(x,y\right)$. The matching process is considered to be completed when the feature points with the greatest correlation are found to correspond to each other through a double-way search. The matching efficiency *r* and Mean Average Precision (MAP) are generally used as two indicators to evaluate the matching effect:(2)$$r=P_{r}/(P_{m}\cdot t)$$
where $p_{r}$ is the number of correct matching feature points extracted from each image, $p_{m}$ is the total number of matching feature points extracted from each image, and *t* is the matching time.
(3)$$MAP=\int_{0}^{1}P\left(R\right)dR$$
where the accuracy rate is $P=p_{r}/p_{m}$, the recall rate is $R=p_{r}/$*p*.

In the case of three different overlapping areas, the Harris and SIFT algorithms are used to match the two adjacent images, and the results are shown in Table 1. According to Formula (3), the average matching accuracy MAP of the SIFT algorithm and Harris algorithm is 0.539 and 0.474, respectively. As the overlap area increases, the matching time (*t*) will also increase correspondingly, and the recall rate (*R*) will gradually increase, resulting in a gradual decrease in the matching efficiency. The matching efficiency of the SIFT algorithm shows more than three times that of Harris algorithm.

Figure 7 (Harris algorithm) and Figure 8 (SIFT algorithm) show the first set of mosaic images in Table 1, respectively. In Figure 7, both of the $S_{12}$ and $S_{34}$ have obvious abnormal matching points and sutures. There are no such circumstances in Figure 8.

In Figure 7, we can see that some matching points far from the stitching position have also been identified by the Harris algorithm(as indicated by the yellow circle), which is obviously incorrect. From the comparison of the results above, it can be seen that the SIFT algorithm has higher computational efficiency and reliability in the stitching of three-dimensional micro-surface morphology images. Therefore, this article uses the SIFT algorithm to implement the stitching of the micro-surface morphology images of parts.

### 2.3. Image Fusion Algorithm

In the image fusion algorithm, $S_{1}\left(x,y,z\right)$ and $S_{2}\left(x,y,z\right)$ are used to represent point sets for images $S_{1}$ and $S_{2}$. $R\left(x,y,z\right)$ represents the common area of both in the mosaic image $S_{12}$, as shown in Figure 9. Since $S_{1}\left(x,y,z\right)$ and $S_{2}\left(x,y,z\right)$ belong to different coordinate systems, $o_{1}-u_{1}v_{1}w_{1}$ and $o_{2}-u_{2}v_{2}w_{2}$, in order to obtain the point set *S*$\left(x,y,z\right)$ of the mosaic image $S_{12}$ in the global coordinate system o-xyz, a coordinate geometric transformation is required.

Image fusion can be divided into pixel-level fusion, feature-level fusion, and decision-level fusion. Since the mosaic images discussed in this paper are the images obtained under the same lens, pixel-level fusion is selected. Pixel-level fusion methods usually include the average fusion method, gradual in and gradual out fusion method, wavelet fusion method, etc.

(1)The average fusion method is used to directly add the corresponding height values of the overlapping areas after matching and take the average value. The calculation formula can be expressed as follows:


(4)
$$S\left(x,y,z\right)=\left\{\begin{array}{c} S_{1}\left(x,y,z\right),\left(x,y\right)\in S_{1}\\ \left(S_{1}\left(x,y,z\right)+S_{2}\left(x,y,z\right)\right)/2,\left(x,y\right)\in R\\ S_{2}\left(x,y,z\right),\left(x,y\right)\in S_{2} \end{array}\right.$$


(2)The gradual in and out method is also known as the linear transition fusion method, and its algorithm can be expressed as follows:
(5)$$S\left(x,y,z\right)=\left\{\begin{array}{c} S_{1}\left(x,y,z\right),\left(x,y\right)\in S_{1}\\ aS_{1}\left(x,y,z\right)+\left(1-a\right)S_{2}\left(x,y,z\right),\left(x,y\right)\in R\\ S_{2}\left(x,y,z\right),\left(x,y\right)\in S_{2} \end{array}\right.$$
where $a=\left(x_{r}-x_{i}\right)/\left(x_{r}-x_{1}\right)$, $x_{1}$ is the abscissa value of the left boundary of the overlapping area. $x_{r}$ is the abscissa value of the right boundary of the overlapping area. $x_{i}$ is the abscissa of pixel points in the overlapping area. *a* is a coefficient between 0 and 1. When a changes from 0 to 1, the two images can smoothly transition.

(3)The wavelet fusion method is used to decompose the overlapping regions corresponding to a group of 3D topography images by a two-dimensional discrete wavelet. The morphology image is decomposed into low-frequency coefficients and high-frequency coefficients in three directions. The algorithm can be expressed by the following formula:
(6)$$S\left(x,y,z\right)=\left\{\begin{array}{c} S_{1}\left(x,y,z\right),\left(x,y\right)\in S_{1}\\ C_{j1}*S_{1}\left(x,y,z\right)+C_{j2}*S_{2}\left(x,y,z\right),\left(x,y\right)\in R\\ S_{2}\left(x,y,z\right),\left(x,y\right)\in S_{2} \end{array}\right.$$
where $C_{j1,2}$ are the decomposition rules for the low-frequency coefficients and high-frequency coefficients, described in detail in the literature [46,47].

Figure 10 shows the fusion results of $S_{1}$ and $S_{2}$ using the above three methods, which is to extract the height values of two different interfaces in the overlapping region $R\left(x,y,z\right)$. It can be concluded that the curve created by the gradual in and out fusion method is the closest to the reality height curve fitting, which represents the actual surface.

### 2.4. Splicing Effect Judgment

This section evaluates the splicing effect by calculating the structural similarity coefficient SSIM of overlapping areas. Given two images *x* and *y*, the SSIM can be expressed as follows:(7)$$SSIM\left(x,y\right)=\left(2\mu_{x}\mu_{y}+c_{1}\right)\left(2\sigma_{xy}+c_{2}\right)/\left(\mu_{x}^{2}+\mu_{x}^{2}+c_{1}\right)\left(\sigma_{x}^{2}+\sigma_{y}^{2}+c_{2}\right)$$
where $\mu_{x}$ and $\mu_{v}$ are the average values of *x* and *y*. $\sigma_{x}$ and $\sigma_{v}$ are the variance of *x* and *y*. $\sigma_{xv}$ is the covariance of *x* and *y*. *L* is the dynamic pixel value. $c_{1}$ and $c_{2}$ are calculated by formulas $c_{1}={\left(k_{1}L\right)}^{2}\;$ and $c_{2}={\left(k_{2}L\right)}^{2}$. The $k_{1}$ and $k_{2}$ are generally selected to be 0.01 and 0.03 [47]. The value range of SSIM can be found in [47,48]. When taking a value of 1, it means that the two images are identical, while the closer to 1, the better the stitching effect.

Formula (10) can be used to calculate the SSIM value of the overlapping area. Comparing the three fusion algorithms described in Section 2.3, the results are shown in Table 2, which proves that the gradual in and out fusion method is the optimal method.

Therefore, the gradual in and out fusion method is used to splice $S_{1}$, $S_{2},{\;S}_{3},{\;\mathrm{and}\;S}_{4}$. The results are shown in Figure 11 and Figure 12, and its 3D shape size is 1281 μm × 1281 μm.

## 3. Wavelet Decomposition and Surface Reconstruction

Spliced images created in the previous steps lack the datum features or roughness parameters necessary for simulation and cannot be saved as solid models to import into simulation software for subsequent analysis. Wavelet analysis separates detection signals, digitally establishing datum planes and roughness surfaces. The pooling method from a neural network algorithm simplifies the shape sample data to reduce calculation requirements and build high-fidelity solid models. The results undergo review to ensure accuracy before evaluating error values through a comparison with actual part surface test data.

### 3.1. Wavelet Decomposition of Detection Signals

The measurement data obtained with the optical profiler are the coordinate values in the *X*, *Y*, and Z directions in the measurement area. If the number of samples in the *X* axis and *Y* axis are recorded as *m* and *n*, respectively, the height matrix of part surface can be expressed as follows:(8)$$Z_{m\times n}\approx D_{m\times n}+R_{m\times n}$$
where $Z_{m\times n}$ is the height matrix of the actual surface topography, $D_{m\times n}$ is the height matrix of the reference plane composed of low-frequency signals such as surface waviness and geometric shape deviation, and $R_{m\times n}$ is the height matrix of high-frequency signals composed of surface roughness.

The wavelet transform method is used to decompose the signal at multiple scales. The wavelet base of the wavelet transform method is a curve with a zero value. By stretching and translating it, a family of wavelet functions is obtained. The signal is decomposed into a linear superposition of the family of wavelet functions [4]. The basic idea of 2D wavelet decomposition is to decompose the original signal *S* into *n* high-frequency signals (*D*_1_, *D*_2_, …, *Dn*) and *n* low-frequency signals (*A*_1_, *A*_2_, …, *An*). Wavelet reconstruction is the inverse process of wavelet decomposition. The reconstructed signal $Z_{m\times n}^{\prime }$ is composed of high-frequency signals at all levels and low-frequency signals at the highest order, which can be expressed as follows:(9)$$Z_{m\times n}^{\prime }=A_{n}+\sum_{i=1}^{n}D_{i},i\in Z$$

The selection of wavelet basis function affects the distribution of wavelet coefficients after decomposition, and directly determines the accuracy of the decomposed signal [4,49]. In this paper, based on the principle of minimum reconstruction deviation, different wavelet bases are used to decompose and reconstruct the original signal, and the Root Mean Square Error (RMSE) of $Z_{m\times n}^{\prime }$ is calculated to find the optimal wavelet base.
(10)$$RMSE=\sqrt{\frac{1}{mn}\sum_{i=1}^{m}\sum_{j=1}^{n}{\left(Z_{i,j}^{\prime }-Z_{i,j}\right)}^{2}},i,j\in Z$$
where $Z_{i,j}^{\prime }$ is the value of row *i* and column *j* in the reconstructed topography height matrix under different wavelet bases, and $Z_{i,j}$ is the value of row *i* and column *j* in the original topography height matrix.

For specific surface data, the decomposition scale *l* is customized to divide the roughness surface and datum plane:(11)$$R_{m\times n}=\sum_{n=1}^{l}D_{i},\;i\in Z$$
where $A_{l}$ is the low-frequency signal coefficient under the decomposition scale *l*, and $\sum_{n=1}^{l}D_{i}$ is the sum of all high-frequency signal coefficients in the *l* to l decomposition scale.

Using *S*_1_ in Figure 4 as an example to compare the Daubechies wavelet, Symlet wavelet, and Coiflet wavelet, the 3D coordinates of some sampling points of *S*_1_ are shown in Table 3. The RMSE results are shown in Table 4 by decomposing and reconstructing the original data with vanishing moments of different orders.

It can be concluded from Table 4 that the optimal wavelet base is sym5, and the minimum reconstruction error is 0.85669 × 10^−19^. Therefore, sym5 is used to decompose the original signal, and the proportion of signal energy at different scales in MATLAB is shown in Figure 13. It can be seen from the figure that the proportion of low-frequency signal energy changes from falling to rising with the increase in decomposition scale *N*, and the minimum value is obtained when *N* = 8. Wavelet decomposition conforms to the law of the conservation of energy. The lower the proportion of low-frequency energy, the closer the low-frequency signal is to the reference plane. Therefore, a Symlet wavelet basis is used to decompose *S*_1_. Under the condition of order 5 and decomposition level 8, the datum plane $D_{m\times n}$ and roughness surface $R_{m\times n}$ obtained are shown in Figure 14.

### 3.2. Simplification and Reconstruction of Surface Model

In order to improve the computational efficiency, the maximum pooling method in the convolution neural network is used to sample and process the shape data. Pooling has translation invariance, which can reduce the difficulty and number of parameters in the optimization when applied to the neural network [5].

Similarly, it can effectively reduce the amount of data and retain the main features of the original topography when processing the surface topography data. For the height matrix m×n of the reconstructed surface, select the pool filter window size of a×a, step size of *a*, and the height matrix m/a×n/a will be obtained by pool processing. If m_1_ = *m/a*, *n*_1_ = *n*/*a*, the simplified shape height matrix $Z_{m_{1}\times n_{1}}^{\prime \prime }$ is obtained. The calculation process is as follows: $z_{1,1}^{\prime \prime }=max\left(z_{i_{1},j_{1}}^{\prime }\right),\;1\leq i_{1}\leq a,1\leq j_{1}\leq a$; $z_{1,2}^{\prime \prime }=max\left(z_{i_{1},j_{2}}^{\prime }\right),\;1\leq i_{1}\leq a,1\leq j_{2}\leq 2a$; $z_{i,j}^{\prime \prime }=max\left(z_{i_{j},j_{j}}^{\prime }\right),\;\left(i-1\right)*a\leq i_{i}\leq ia,\left(j-1\right)*a+1\leq j_{j}\leq ja$; $z_{m_{1},n_{1}}^{\prime \prime }=max\left(z_{i_{m_{1}},j_{n_{1}}}^{\prime }\right),\;\left(m_{1}-1\right)*a+1\leq i_{m_{1}}\leq m,\;\left(n_{1}-1\right)*a+1\leq j_{n_{1}}\leq n$, where $Z_{i,j}^{\prime \prime }$ is the value of the *i*th row and the *j*th column of the surface topography height matrix $Z_{m_{1}\times n_{1}}^{\prime \prime }$ after down-sampling, and the point cloud model $D_{m_{1}\times n_{1}}^{\prime \prime }$ of the datum plane and $R_{m_{1}\times n_{1}}^{\prime \prime }$ of the rough surface can be obtained by using the same processing method.

Take *S*_1_ in Figure 4 as an example; the reconstructed surface model of *S*_1_ can be obtained by overlying the datum and roughness surfaces in Figure 14. The result is shown in Figure 15a, which has the height matrix m×n of 1024 × 1024. If *a* = 4 is taken for sampling processing, then the simplified height matrix is 256 × 256, which can be processed in MATLAB, as shown in Figure 15b.

The discrete point cloud model is transformed into a continuous geometric model by the reverse modeling method, which include explicit and implicit reconstruction methods. The explicit method triangulates the point cloud model directly, while the implicit method obtains the final triangulation and extracts the isosurface of the function by reconstructing the implicit function. The Poisson surface reconstruction used in this paper is established by Meshlab, which is detailed in Appendix B. Figure 16c shows the high-fidelity surface model after six times magnification, which is a combination of Figure 16a,b. It can be directly used in the simulation of the machining process. 

### 3.3. Surface Model Deviation Judgement

Deviation detection is an important basis for judging whether the reconstructed surface model is consistent with the actual topography. The surface roughness height parameters $S_{a}$,$\;S_{q},$ and $S_{z}$ represent the arithmetic mean height, root mean square height, and maximum height of the three-dimensional surface topography, which depend on the surface height deviation. To verify the precision of the high-fidelity surface model, we compare it with the measured surface and calculate the relative deviation. The calculation process is as follows: $\Delta_{1}=\left|S_{a_{m}}-S_{a_{h}}\right|/S_{a_{m}}\times 100\%\leq e$; $\Delta_{2}=\left|S_{q_{m}}-S_{q_{h}}\right|/S_{q_{m}}\times 100\%\leq e$; $\Delta_{3}=\left|S_{z_{m}}-S_{z_{h}}\right|/S_{z_{m}}\times 100\%\leq e$, where, $\Delta_{1}$, $\;\Delta_{2}$, and $\Delta_{3}$ are relative deviations; $S_{a_{m}}$, $\;S_{q_{m}}$, and $\;S_{z_{m}}$ are calculated based on the actual part surface measurement; $S_{a_{h}},\;\;S_{q_{h}},\;\mathrm{and}\;\;S_{z_{h}}$ are calculated based on the high-fidelity surface model; and *e* is the threshold.

In this paper, the ZeGage Plus optical profiler is used to obtain the $S_{a}$, $S_{q},$ and $S_{z}$ of the part, as shown in Table 5. The actual roughness values of the four sample areas are shown in Figure 17.

Table 6 shows the $S_{a}$, $S_{q},$ and $S_{z}$ values of the four sampling positions ($S_{1},S_{2},S_{3},S_{4}$) in Figure 4, and compares them with the actual measured values.

Table 6 demonstrates that the maximum relative error obtained at the four sampling positions is 0.909 μm. In general, the allowable error range for finish machining is within several micrometers. This proves that this method can meet the high-fidelity modeling requirements of finish machined parts within the allowable error range of 1 μm. Simultaneously, by reconstructing a high-fidelity model after applying wavelet decomposition, the number of grids in the model is significantly reduced, thereby decreasing the computational burden of subsequent machining simulations.

## 4. Comparison with Experiments

To verify the advantages of the high-fidelity model in the machining simulation, the example in Figure 4 was used for the surface grinding experiment. The discrete element simulation software EDEM2021 is used to simulate the grinding process. The experimental equipment is a BJL-LL05 vertical centrifugal roller polishing machine (shown in Figure 18a). Its working principle is to place the parts, grinding blocks, grinding agents, water, etc., into four evenly distributed cylinders. The cylinder moves in planetary motion. The revolution speed is *N*, the rotation speed is *n*, and the speed ratio is *n*/*N*. Through the motion created, the grinding block collides, rolls, and micro-grinds the part surface to achieve the finishing of the part surface, as shown in Figure 18b. The roller and installation position of the parts are shown in Figure 18c. Figure 18d is the simulation picture of the grinding experiment in the EDEM simulation.

### 4.1. Grinding Simulation in EDEM

To simulate roller grinding, four main parameters need to be considered: material parameters, contact parameters, grinding blocks factory parameters, and geometry parameters. The material parameters, as listed in Table 7, comprise the Poisson’s ratio, density, and shear modulus of the rolling grinding blocks, part, and roller.

Contact parameters, displayed in Table 8, encompass the rebound coefficient, static friction coefficient, and rolling friction coefficient between the grinding blocks, part, and roller.

To define the grinding blocks used in the simulation, a spherical block with a diameter of 3 mm, a filling rate of 70%, and a total of 22,000 blocks were specified as the grinding block factory parameters. Figure 19 illustrates the interface of the grinding blocks factory in EDEM2021 software, which includes the blocks mass, volume, and rotational inertia in X, Y, and Z directions.

The revolution speed of the roller is 300 r/min, the revolution radius is 135 mm, and the direction is clockwise. The rotation speed of the drum is 300 r/min and the direction is counterclockwise. The simulation time is 60 s, within which the roller makes planetary motion. The movement track of the roller in one cycle is shown in Figure 20. The time step is set to 20% Rayleigh time, and the calculated time step ∆*t* = 1.5 × 10^−5^ s.

In the simulation experiment, the wear depth nephogram with a time interval of 10 s is selected to analyze the wear condition of the parts. The wear condition of the ideal surface model and the high-fidelity surface model in the EDEM simulation is shown in Figure 21 and Figure 22. It can be seen from the wear nephogram of the common model that the scratches on the surface of the parts are disorderly. However, the high-fidelity model shows regular wear traces.

### 4.2. Physical Experiment Results

After 5 min of machining, the surface roughness $S_{a}$ of $S_{1}$ changes from 1.272 μm reduced to 0.565 μm. The $S_{q}$ of $S_{1}$ is reduced from 1.624 μm to 0.708 μm. The $S_{z}$ of $S_{1}$ is reduced from 15.262 μm to 6.162 μm. Figure 23 is used to compare the wear nephogram of the ideal model and the high-fidelity model in the EDEM simulation with the image of the parts after actual grinding.

In light of the excessive computational complexity, the experimental results were solely compared after 60 s of simulation processing. The similarity between Figure 23a,c denoted as *SIM*_1_ and *SIM*_2_, represents the similarity between Figure 23b,c. The similarity value is obtained by calculating the normalized correlation coefficient of the two images [22], as shown in Formula (12).
(12)$$SIM=\sum_{m}\sum_{n}\left(A_{mn}-\bar{A}\right)\left(B_{mn}-\bar{B}\right)/\sqrt{\left(\sum_{m}\sum_{n}{\left(A_{mn}-\bar{A}\right)}^{2}\right)\left(\sum_{m}\sum_{n}{\left(B_{mn}-\bar{B}\right)}^{2}\right)}$$
where *A_mn_* and *B_mn_* are the value of the *m*th row and *n*th column in the gray value matrix *A* and *B* of the two images. $\bar{A}$ and $\bar{B}$ are the average of the pixels of the matrix.

Using Formula (12), *SIM*_1_ = −0.0237 and *SIM*_2_ = 0.3093 are obtained. The experimental comparison proved that this method is an improvement on current simulation accuracy by more than 30%. That is, compared with the ideal model, the high-fidelity model is more in line with the wear situation in the actual part surface.

## 5. Conclusions and Future Work

This paper proposed a novel method for developing a high-fidelity surface model. The experiments show that the simulation accuracy can be improved by more than 30% using the high-fidelity model compared with the ideal model. The method of the image mosaic can effectively compensate for the shortcomings of the measuring lens. It makes it possible to obtain a wide range of high-precision part surface panoramas. Some deficiencies of the paper still need further study. It can be seen from the comparison of simulation results in Figure 23 that although the result of SIM2 is much higher than that of *SIM*_1_, it also took 96.708 h to complete this simulation, which was undertaken by a computer with an Intel i7-10700 processor (Intel corporation, Santa Clara, CA, USA), a main frequency of 2.9 GHz, a 32 GB memory, and a 16-core CPU.

In the field of high-end manufacturing, high-precision CNC machining machines have demonstrated remarkable accuracy, often achieving tolerances as low as 0.001 mm. Industries such as MEMS (Micro-Electro-Mechanical Systems) and optical component manufacturing demand even greater levels of precision, necessitating a surface machining accuracy at the micron or sub-micron level. Consequently, the development of high-fidelity models is crucial to meet these requirements. However, when analyzing the data collected from optical measurements, discrepancies arise due to variations in the results obtained through different filtering methods. Surprisingly, the existing literature rarely addresses the effectiveness of utilizing these high-fidelity models in machining simulations. This article aims to address this gap by providing valuable insights and supplementary experiments on this important topic. In future work, we will try to use different mathematical statistical methods to simplify the sampling of the data, and in this way, reduce the calculations. Moreover, with the improvement of computer computing capacity, the method proposed in this paper is expected to be more easily applied in practical applications.

## Figures and Tables

**Figure 1 micromachines-15-00834-f001:**
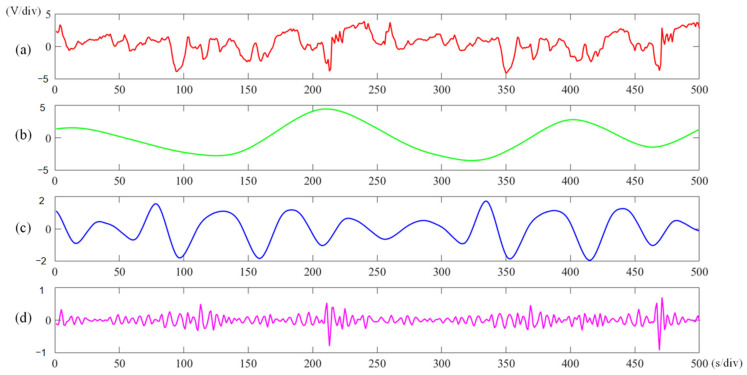
Schematic diagram of multi-scale characteristics of surface topography of (**a**) composite surface feature; (**b**) shape deviation; (**c**) surface waviness; and (**d**) surface roughness.

**Figure 2 micromachines-15-00834-f002:**
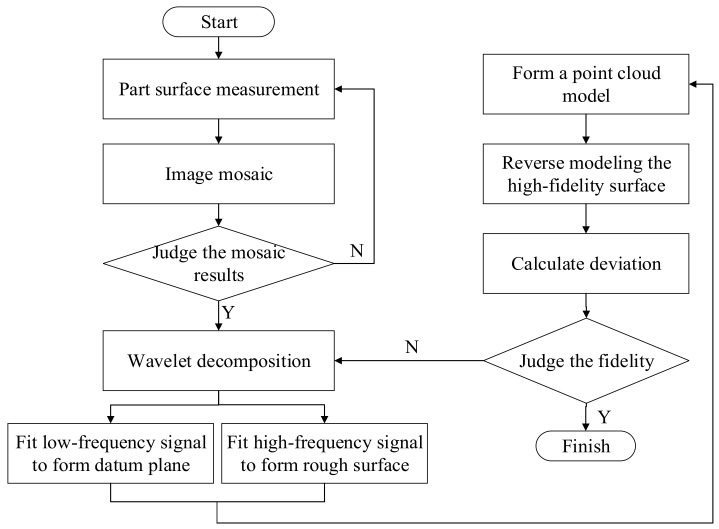
High-fidelity surface modeling flow chart.

**Figure 3 micromachines-15-00834-f003:**
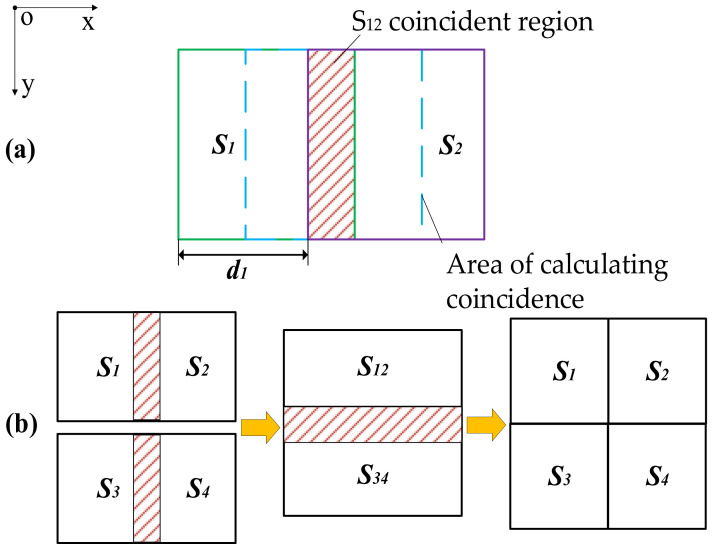
Schematic diagram of image acquisition and mosaic of (**a**) acquisition process; (**b**) stitching process.

**Figure 4 micromachines-15-00834-f004:**
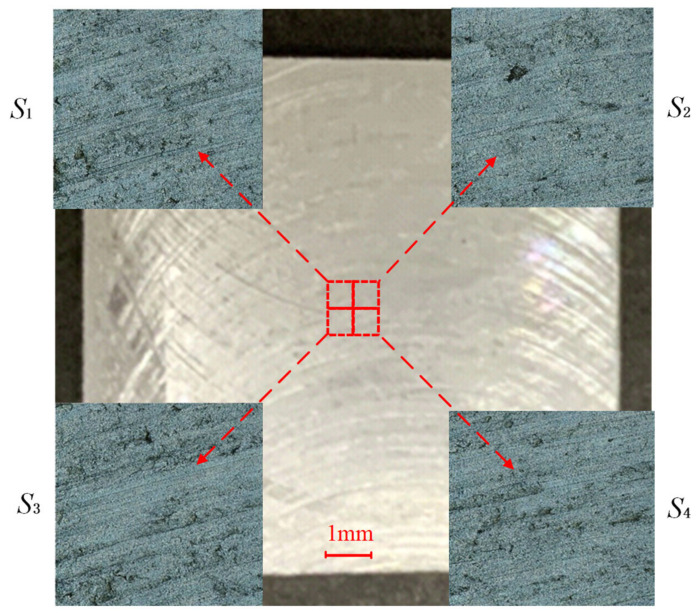
Micro morphology of sampling area.

**Figure 5 micromachines-15-00834-f005:**
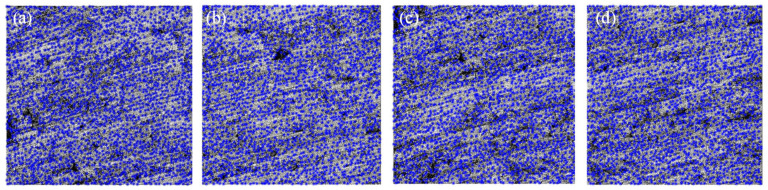
Test results of Harris corner detection of (**a**) *S*_1_; (**b**) *S*_2_; (**c**) *S*_3_; and (**d**) *S*_4_.

**Figure 6 micromachines-15-00834-f006:**
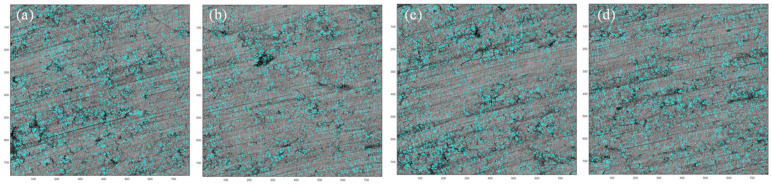
Test results of SIFT feature descriptor detection of (**a**) *S*_1_; (**b**) *S*_2_; (**c**) *S*_3_; and (**d**) *S*_4_.

**Figure 7 micromachines-15-00834-f007:**
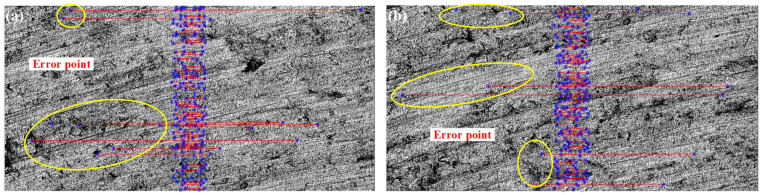
Matching point connection of Harris algorithm of (**a**) Harris feature point matching of *S*_1_, *S*_2_; (**b**) Harris feature point matching of *S*_3_, *S*_4_.

**Figure 8 micromachines-15-00834-f008:**
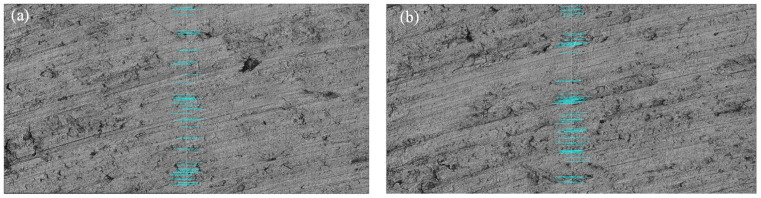
Matching point connection of SIFT algorithm of (**a**) SIFT feature point matching of *S*_1_, *S*_2_; (**b**) SIFT feature point matching of *S*_3_, *S*_4_.

**Figure 9 micromachines-15-00834-f009:**
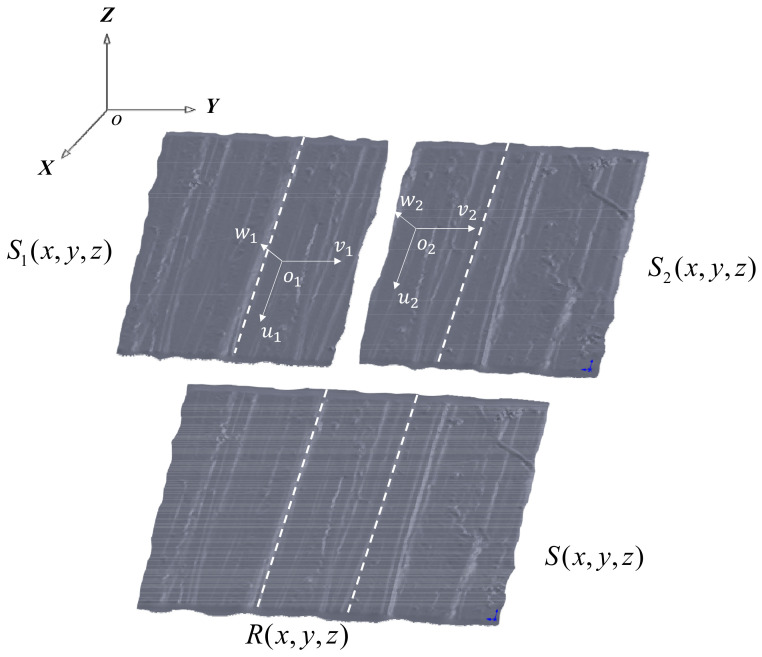
Schematic diagram of three-dimensional topography splicing.

**Figure 10 micromachines-15-00834-f010:**
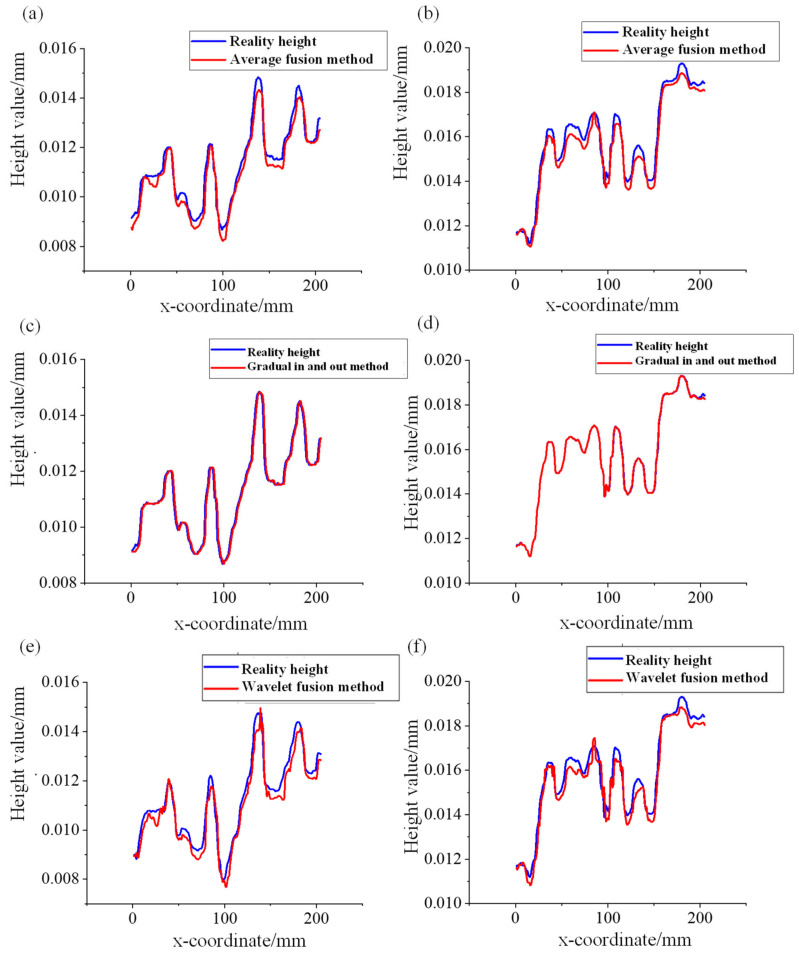
Height curves of different image fusion methods at different sections of (**a**) average fusion method for $S_{1}$; (**b**) gradual fusion method for $S_{1}$; (**c**) wavelet fusion method for $S_{1}$; (**d**) average fusion method for $S_{2}$; (**e**) gradual fusion method for $S_{2}$; and (**f**) wavelet fusion method for $S_{2}$.

**Figure 11 micromachines-15-00834-f011:**
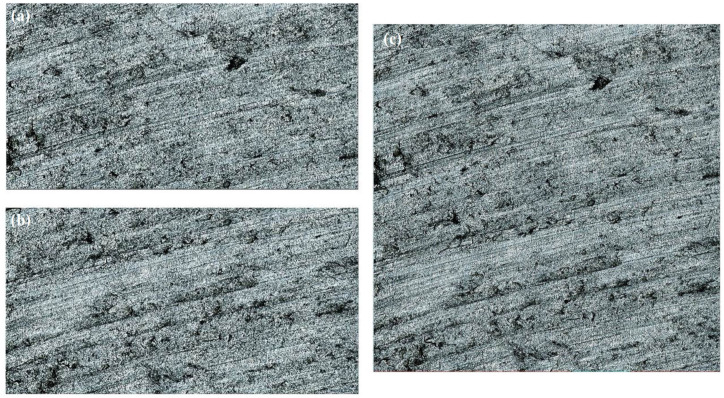
Image fusion effects: (**a**) *S*_12_ is spliced by *S*_1_, *S*_2_; (**b**) *S*_34_ is spliced by *S*_3_, *S*_4_; and (**c**) *S* is spliced by *S*_12_, *S*_34._

**Figure 12 micromachines-15-00834-f012:**
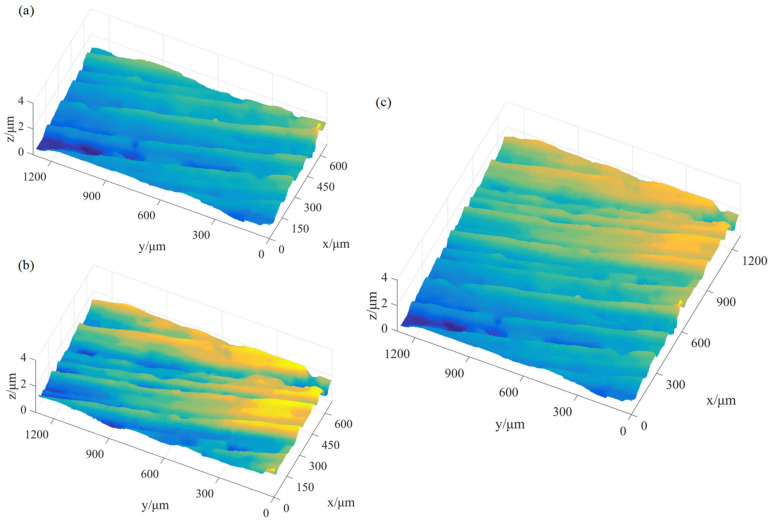
Three-dimensional morphologies of spliced images: (**a**) *S*_12_ is spliced by *S*_1_, *S*_2_; (**b**) *S*_34_ is spliced by *S*_3_, *S*_4_; and (**c**) *S* is spliced by *S*_12_, *S*_34_.

**Figure 13 micromachines-15-00834-f013:**
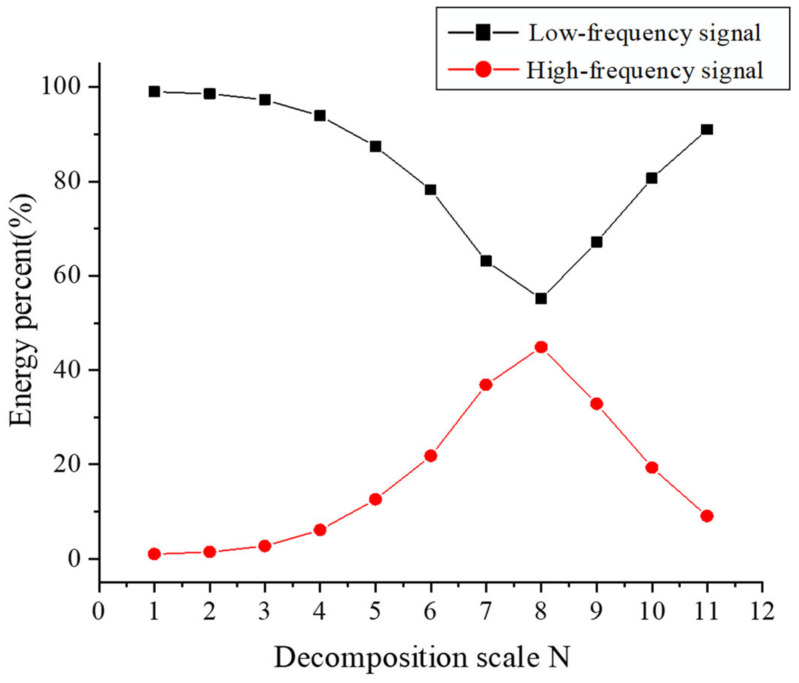
Energy proportion diagram at different decomposition scales.

**Figure 14 micromachines-15-00834-f014:**
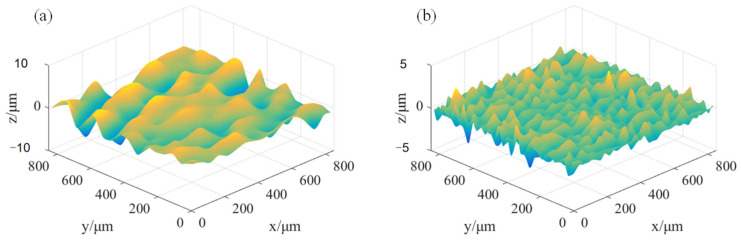
The surface of different morphologies at position *S*_1_ of the specimen of (**a**) datum surface; (**b**) roughness surface.

**Figure 15 micromachines-15-00834-f015:**
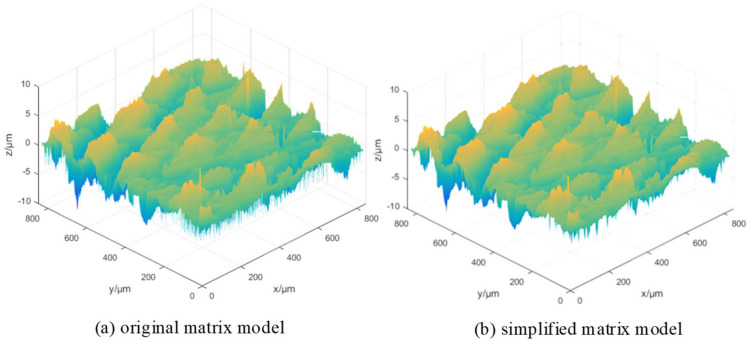
Signal comparison diagram before and after the down-sampling.

**Figure 16 micromachines-15-00834-f016:**
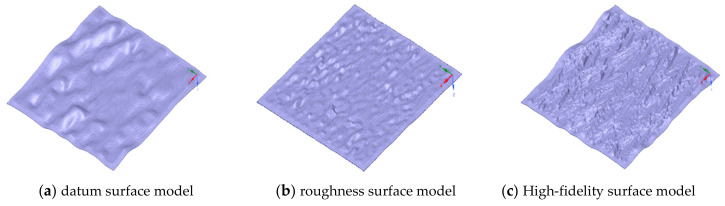
Surface geometry model of specimen.

**Figure 17 micromachines-15-00834-f017:**
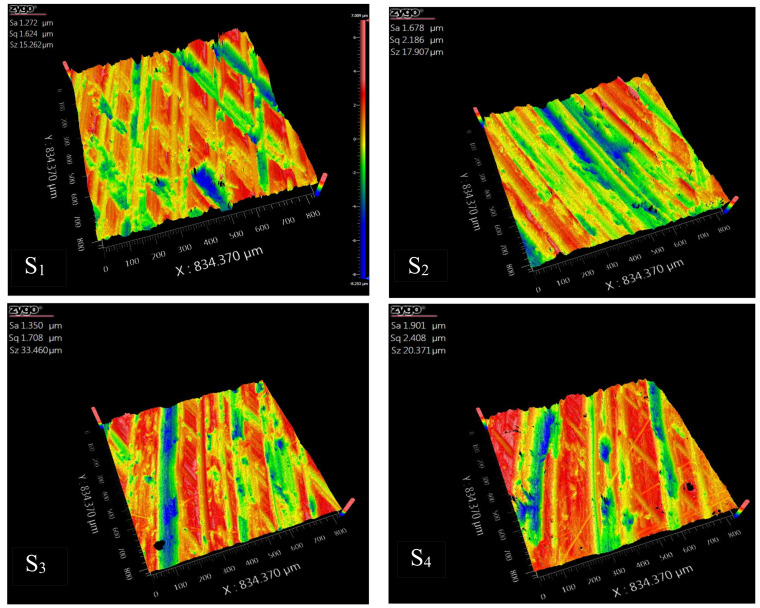
Part surface images before finishing.

**Figure 18 micromachines-15-00834-f018:**
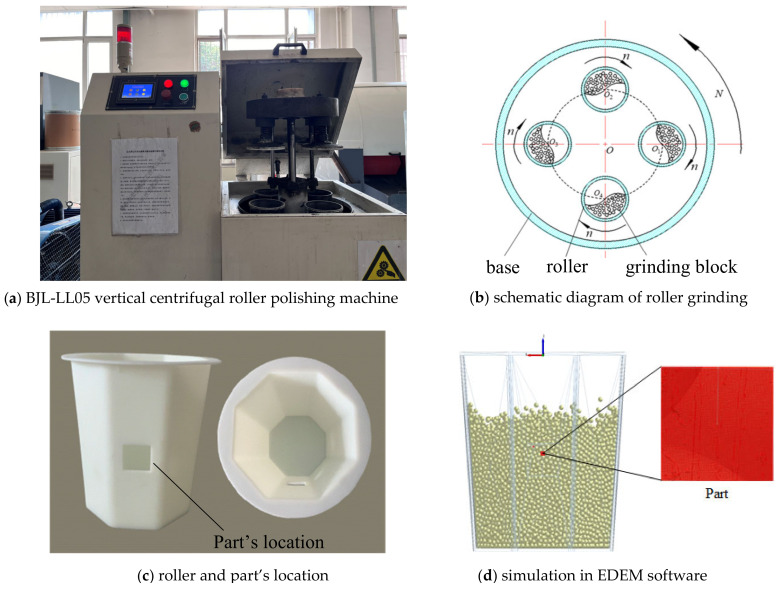
Experiment and simulation of part surface grinding.

**Figure 19 micromachines-15-00834-f019:**
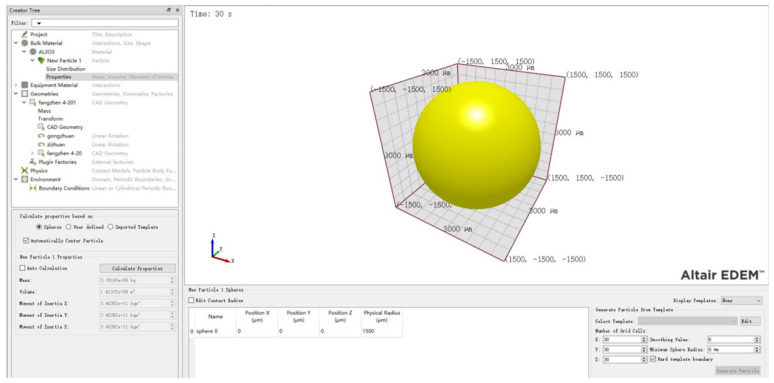
Particle property parameters.

**Figure 20 micromachines-15-00834-f020:**
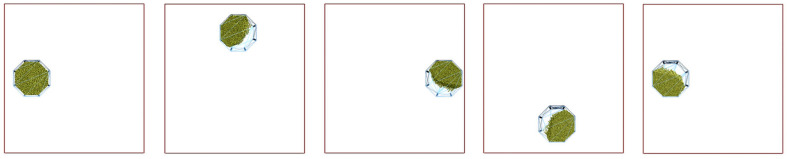
The trajectory of the roller in a period.

**Figure 21 micromachines-15-00834-f021:**
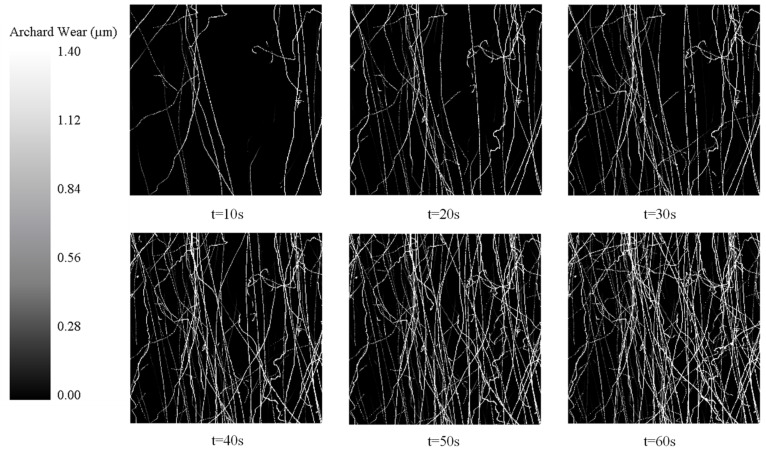
Wear depth cloud image of ideal surface model.

**Figure 22 micromachines-15-00834-f022:**
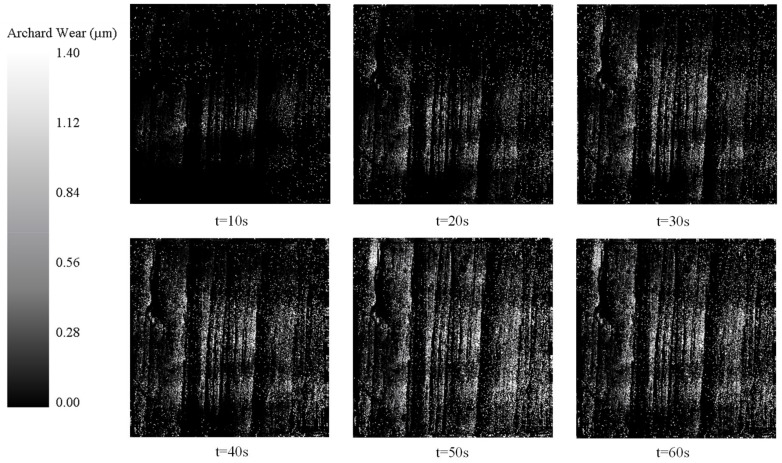
Wear depth cloud map of high-fidelity surface model.

**Figure 23 micromachines-15-00834-f023:**

Comparison of wear nephogram of simulation models and actual part.

**Table 1 micromachines-15-00834-t001:** Algorithm matching results.

NO.	Image	Algorithm	Feature Points *p*	*p_m_*	*p_r_*	*t* (*s*)	*r*	*P*	*R*
1 (10% overlapping area)	*S*_1_, *S*_2_	Harris	3112	100	92	15.92	0.058	0.920	0.030
		SIFT	3009	83	83	4.76	0.210	1.000	0.028
	*S*_3_, *S*_4_	Harris	3160	106	100	14.18	0.067	0.943	0.032
		SIFT	3218	98	98	5.12	0.195	1.000	0.030
2 (20% overlapping area)	*S*_1_, *S*_2_	Harris	3506	240	237	18.92	0.052	0.988	0.068
		SIFT	3367	258	258	5.34	0.187	1.000	0.077
	*S*_3_, *S*_4_	Harris	3521	241	239	19.87	0.050	0.992	0.068
		SIFT	3578	291	291	5.71	0.175	1.000	0.081
3 (30% overlapping area)	*S*_1_, *S*_2_	Harris	4331	605	605	25.98	0.038	1.000	0.140
		SIFT	4186	679	679	6.70	0.149	1.000	0.162
	*S*_3_, *S*_4_	Harris	4330	614	613	26.20	0.038	0.998	0.142
		SIFT	4304	693	693	6.97	0.144	1.000	0.161

**Table 2 micromachines-15-00834-t002:** Evaluation of splicing effect under different fusion methods.

Images	Methods	SSIM
*S*_1_, *S*_2_	The average fusion method	0.927
	The gradual in and out method	0.979
	The wavelet fusion method	0.933
*S*_3_, *S*_4_	The average fusion method	0.937
	The gradual in and out method	0.980
	The wavelet fusion method	0.937
*S*_12_, *S*_34_	The average fusion method	0.917
	The gradual in and out method	0.977
	The wavelet fusion method	0.928

**Table 3 micromachines-15-00834-t003:** Part of 3D coordinate data of surface topography at the position *S*_1_ of specimen.

*Z-Axis*/μm	*X-Axis*/μm	*Y-Axis*/μm
2.3635	0	0
2.3177	0.8148	0
2.28	1.6296	0
2.2279	2.4444	0
2.3975	0	0.8148
2.3824	0.8148	0.8148
2.3354	1.6296	0.8148
2.2645	2.4444	0.8148
···	···	···
0.1451	832.7409	834.3706
0.1949	833.5557	834.3706
0.2917	834.3706	834.3706

**Table 4 micromachines-15-00834-t004:** RMSE values of different wavelet bases.

	Wavelet Basis	RMSE (×10^−19^)
Daubechies	db2	2.6299
	db3	32.379
	db4	5.8918
	db5	9.7226
	db6	6.5075
Symlet	sym4	2.3910
	sym5	0.85669
	sym7	4.0252
	sym8	1.1052
Coiflet	coif2	46.183
	coif3	2.5555
	coif4	124.44

**Table 5 micromachines-15-00834-t005:** ZeGagage Plus main performance parameters.

Equipment	Performance	Value
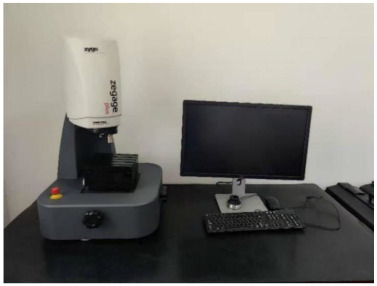	Magnification	1–50
	Measurement array	1024 × 1024, 512 × 512, 256 × 256, 1024 × 160
	Transverse scan travel	100 × 100 mm
	Longitudinal scanning stroke	≤20 mm
	Surface topography	<0.15 nm
	RMS repeatability	0.01 nm
	Optical resolution	0.52 μm
	Scanning speed	≤73 μm/s

**Table 6 micromachines-15-00834-t006:** $S_{a}$, $S_{q}$ and $S_{z}$ value error between simulation morphology and actual morphology of specimen at different sampling positions.

Sample Location		*S_a_*/μm	*S_q_*/μm	*S_Z_*/μm	Maximum Relative Error
$S_{1}$	Part surface	1.272	1.624	15.262	0.659
	High-fidelity surface	1.280	1.633	14.603	
	Relative error	0.08	0.009	0.659	
$S_{2}$	Part surface	1.678	2.186	17.907	0.157
	High-fidelity surface	1.672	2.180	17.750	
	Relative error	0.006	0.006	0.157	
$S_{3}$	Part surface	1.350	1.708	33.460	0.909
	High-fidelity surface	1.358	1.715	32.551	
	Relative error	0.008	0.007	0.909	
$S_{4}$	Part surface	1.901	2.408	20.371	0.236
	High-fidelity surface	1.889	2.398	20.135	
	Relative error	0.012	0.010	0.236	

**Table 7 micromachines-15-00834-t007:** Material properties in EDEM simulation.

	Grinding Block	Part	Roller
Material	Al_2_O_3_	Aluminium alloy	Photosensitive resin
Poisson ratio	0.36	0.33	0.4
Elastic modulus/Pa	1.26 × 10^7^	2.632 × 10^10^	9.246 × 10^8^
Density/(kg∙m^−3^)	2675	2700	1150

**Table 8 micromachines-15-00834-t008:** Contact parameters.

	Grinding Blocks	Part-Grinding Block	Roller-Grinding Block
Rebound coefficient	0.35	0.5	0.35
Static coefficient	0.15	0.45	0.3
Dynamic coefficient	0.46	0.15	0.15

## Data Availability

The latest experimental data can be found at https://www.researchgate.net/profile/Yupeng-Xin, accessed on 20 May 2024. More new data is unavailable due to privacy restrictions.

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
