# Peer review of "A Novel High-Fidelity Simulation for Finishing Operations: Hybrid Image Mosaic and Wavelet Decomposition"

_micromachines, 2024, doi:10.3390/mi15070834_

Round 1

Reviewer 1 Report

Comments and Suggestions for Authors

This is an interesting paper and deserves to be published. 

The authors might consider literature from laboratories at universities in Halmstad, Huddersfield, Nottingham, Valenciennes, Cachan, and Poznan.

The authors should look at national and international standards like, ISO 25178 and ASME B46.1 and adjust terms to be consistent with the standards. Note that the verb "to form" could be replaces with "to create" or "to produce". Surface topographies can be decomposed, according to standards into form. waviness, and roughness. Using form as a verb can be confusing. Note how sampling is used in the standards. Heights are sampled at locations separated by sampling intervals.

Comments on the Quality of English Language

adequate

Reviewer 2 Report

Comments and Suggestions for Authors

It is meaningful for the authors to propose a novel high-fidelity modeling method combining image mosaic and wavelet decomposition technologies. There are problems should be noticed as following:

(1) All parameters in the text and equations should be italic.

(2) Figures 17 is not very clear. The axis need to be given clearly. Is it 3D or 2D data

(3) Figure 22 is not very clear.

(4) The figure 19 is not very related to the surface images splicing. The comparison experiments are better to comparing with the previous surface images splicing.

(5) It suggests the wear surface is also be measured by ZeGagage Plus surface topography.

Comments on the Quality of English Language

The language do not need to revise. 
